# Performance of Reconfigurable-Intelligent-Surface-Assisted Satellite Quasi-Stationary Aircraft–Terrestrial Laser Communication System

Yi Wang [1,2,*], Haibo Wang [1] and XueWen Jiang [1]

1   Key Laboratory of Electromagnetic Wave Information Technology and Metrology of Zhejiang Province, College of Information Engineering, China Jiliang University, Hangzhou 310018, China
2   State Key Laboratory of Oil and Gas Reservoir Geology and Exploitation, Southwest Petroleum University, Chengdu 610500, China
*   Correspondence: wcy16@cjlu.edu.cn

**Abstract:** This paper proposes the use of quasi-stationary aircraft and reconfigurable intelligent surfaces (RIS) to improve the system performance in satellite–terrestrial laser communication downlink. Single-input multiple-output (SIMO) technology is applied to the relay node of a quasi-stationary aircraft. The closed expression of the bit error rate (BER) of an RIS-assisted satellite quasi-stationary aircraft–terrestrial laser communication system (RIS-SHTLC) is derived under the M-distributed atmospheric turbulence model while considering the influence of atmospheric turbulence and pointing errors caused by RIS jitter. The effects of coherent binary frequency shift keying (CBFSK), coherent binary phase-shift keying (CBPSK), non-coherent binary frequency shift keying (NBFSK), and differential binary phase-shift keying (DBPSK) on the performance of an RIS-SHTLC system are simulated and analyzed under weak turbulence. The results show that the RIS-SHTLC system with CBPSK modulation has the best communication performance. Simultaneously, the relationships between the average signal-to-noise ratio (SNR) and BER of the RIS-SHTLC system under different RIS elements are simulated and analyzed, and compared with the traditional SHTLC system. In addition, the influence of the zenith angle, receiving aperture and divergence angle on the performance of the system is studied. Finally, Monte Carlo simulations are used to validate the analytical results.

**Keywords:** satellite–terrestrial laser communication; downlink; quasi-stationary aircraft; reconfigurable intelligent surface; single-input multiple-output; coherent binary phase-shift keying

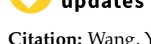



## 1. Introduction

Interest in the use of free-space optical links in satellite communication scenarios has been increasing in recent years. Compared to radio frequency (RF) communications, satellite laser communications have the advantages of a greater bandwidth, higher channel capacity, no spectrum licensing requirement, and better security [1–3]. However, when the optical signal passes through the atmospheric channel, its diameter is considerably larger than the atmospheric vortex size. Consequently, multiple independent atmospheric vortices will appear on the cross-section of the beam, and these vortices will create the independent diffraction and scattering of the beam. This phenomenon causes the beam received by the receiver to exhibit random energy fluctuations, i.e., the light intensity scintillation effect, which can seriously affect the performance of the entire communication system [4]. In addition, pointing errors due to the misalignment of the transmitter and receiver constitute another important factor that affects the performance of the communication system [5]. Various techniques have been proposed to alleviate the impact of these issues on the performance of satellite–terrestrial laser communication systems, such as the modulation technique, multiple-input and multiple-output technique, and relaying technique [6–8].

The satellite–terrestrial laser communication link can also be obstructed by clouds. Therefore, the availability of the downlink depends on the cloud conditions above the terrestrial receiver. A promising solution is the use of a high altitude platform (HAP), which is a quasi-stationary aircraft located at a cloud-free atmospheric altitude of about 20 km above the Earth's surface [9]. Using the HAP as a relay node for satellite–terrestrial laser communication also helps to improve the performance of the communication system [10]. Due to these advantages, studies on the use of HAP as a relay node to assist satellite–terrestrial laser communication have emerged in recent years [10–12]. For example, Vu et al. derived the closed expression of the bit error rate (BER) for the downlink HAP relay system corresponding to satellite–terrestrial laser communication under the Gamma–Gamma atmospheric channel model. The authors found that the transmit power of the HAP-relay-assisted satellite–terrestrial laser communication system with the same BER is 25 dB lower than that of the system without the relay technique [10].

Dang et al. introduced the HAP as a relay node into a satellite–terrestrial laser quantum key distribution system, and derived the ergodic secret-key rate of the system under the Gamma–Gamma atmospheric channel model [11]. Shah et al. considered two scenarios for the satellite–terrestrial laser communication uplink: no HAP deployment and HAP deployment. The authors used the RF link as a backup for the laser link, and derived a unified expression for the outage probability (OP) and BER for this single-hop system and the two-hop (with HAP) system under the Gamma–Gamma atmospheric channel model [12]. However, the above-mentioned literature on satellite-to-HAP links uses the single-input single-output (SISO) technique. Unlike the SISO technique, the single-input multiple-output (SIMO) technique uses multiple receiving apertures to form multiple independent channels, and the diversity gain provided by these multiple independent channels can effectively combat channel fading caused by atmospheric turbulence. To the best of our knowledge, the use of the M distribution in satellite–high altitude platform–terrestrial laser communication (SHTLC) systems and SIMO technology for the satellite-to-HAP link has not been reported in existing literature.

Reconfigurable intelligent surfaces (RIS) can passively reflect signals without any transmitting equipment, and it can also be seen as an alternative to active relay technology [13]. However, compared to the relay technology, the RIS consumes less power, is not affected by the receiver noise at the relay, and does not require complex processing at the relay. The RIS is a planar array composed of multiple mirrors or optical phased array structures, which can intelligently control the amplitude, phase, and polarization of the incident signal through integrated electronics [14]. Based on the aforementioned features, it can effectively customize the wireless environment to enhance the intelligence of the wireless communication channel, thus improving the capacity, spectrum, and energy efficiency of the communication network [15]. RIS has been proposed in recent years in RF communications to solve the dead zone problem in RF networks and to create smarter communication channels [16,17]. In the physical-layer security scheme, RIS-assisted secure communication has also attracted wide attention [18]. The RIS concept had also been expanded to include optical and hybrid systems. Abumarshoud et al. combined RIS and a Light Fidelity (LiFi) transmitter in visible light communication, so as to realize the dynamic tunability of the LiFi transceiver [19]. Yuan et al. used RIS with an RF signal generator as a terminal for signal enhancement in a hybrid RF/FSO unmanned aerial vehicle (UAV) communication system [20].

In the first study on RIS technology in a terrestrial free-space optical (FSO) communication system, Najafi et al. proposed an RIS-assisted FSO system using phase-shift profiles. The geometric and misalignment losses (GML) were modeled to characterize the effect of different RIS physical parameters on the channel. The probability density function (PDF) of the GML was derived for two-dimensional and three-dimensional scenarios, and finally the outage performance of the RIS-assisted FSO system was studied, considering GML and atmospheric turbulence [21]. Different techniques for designing reflective surfaces at optical frequencies were reviewed by Jamali et al. and compared optical IRS with RF IRS

and optical relaying [22]. In [23], the authors analyzed the performance of RIS-assisted terrestrial FSO subjected to turbulence and pointing errors, considering the link distance and RIS jitter ratios at the RIS position. In [24], the authors unified the Fisher–Snedecor (F), Gamma–Gamma (GG) and Málaga (M) distribution atmospheric turbulence models. Closed expressions for the outage probability, BER and channel capacity of the RIS-FSO system are derived considering the effects of atmospheric turbulence, pointing errors, and random fog. In [25], considering the imperfect channel state information (CSI) and pointing errors, the authors derived closed expressions for the PDF and CDF of the RIS-assisted FSO system under the F distribution. The study of RIS in the above article was conducted in a horizontal link FSO system, where the purpose of using the RIS was to relax the line-of-sight requirements of the FSO system. The RIS technology can also be used to improve system performance by using RIS modules with multiple RIS elements to reflect the incident beam. This results in the formation of multiple independent channels to mitigate the effects of atmospheric turbulence, and intelligently control the direction of the reflected beam to reduce the pointing errors. To the best of our knowledge, the use of RIS technology to enhance the performance of vertical link satellite–terrestrial laser communication systems has not been reported in existing literature. In short, the differences between this article and related articles are shown in Table 1.

**Table 1.** Differences between this article and related articles.

| Articles | FSO Terrestrial Communication | Satellite–Laser Communication | HAP | RIS | Multiple-Elements RIS | RIS Jitter |
|---|---|---|---|---|---|---|
| [10] | | √ | √ | | | |
| [11] | | √ | √ | | | |
| [12] | | √ | √ | | | |
| [21] | √ | | | √ | | √ |
| [23] | √ | | | √ | | |
| [24] | √ | | | √ | √ | |
| [25] | √ | | | √ | | |
| This | | √ | √ | √ | √ | √ |

Based on the above analysis, this paper introduces RIS technology into the satellite–HAP–terrestrial laser communication system. The M distribution is used to characterize the atmospheric channel model, and the SIMO technology is used to improve the system performance in the satellite-to-HAP system. Considering the effects of light intensity scintillation and pointing errors, the closed expressions of the PDF and the cumulative distribution function (CDF) for the satellite-to-HAP link and the HAP-to-terrestrial link are derived, respectively.

These derivations are further used to obtain the closed expression for the BER of the RIS-SHTLC system. The effects of CBFSK, CBPSK, NBFSK, and DBPSK on the RIS-SHTLC system performance are simulated and analyzed under weak turbulence. At the same time, the relationship between the average SNR and BER of the RIS-SHTLC system under different RIS elements is simulated and analyzed, and compared with the traditional SHTLC system. Furthermore, the influence of the zenith angle, receiving aperture and divergence angle on the performance of the system is studied. Finally, Monte Carlo simulations are used to verify the correctness of the analytical results.

The rest of this paper is organized as follows: In Section 2, the system model and channel model are presented. In Section 3, the BER of the RIS-SHTLC system is analyzed. The performance of the RIS-SHTLC system is simulated and analyzed and, compared with the SHTLC system in Section 4. The paper is concluded in Section 5.

## 2. System and Channel Model

Figure 1 shows the schematic diagram of the RIS-SHTLC system. In the downlink, the satellite transmits information to the terrestrial receiver with the help of the HAP and RIS. The satellite node is equipped with a laser. The HAP relay node with a multi-aperture

receiver adopts a decode-and-forward relay protocol (DF). The RIS module has N reflective elements, and the terrestrial receiver has a single aperture.

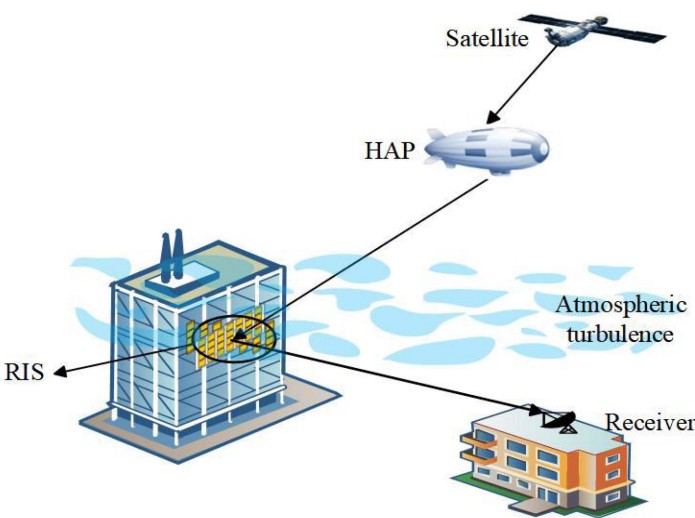

**Figure 1.** Schematic diagram of the RIS-SHTLC system.

### 2.1. Satellite-to-HAP Link

In the satellite-to-HAP link, the HAP receiver uses equal gain combination (EGC) to combine signals, i.e., assigning equal weights to the signals on each receive aperture. The advantage of EGC over other merging techniques is that it does not require channel state information. The output SNR of the HAP with EGC can be expressed as [26]

$$\gamma_{s-p} = \overline{\gamma}_{s-p} \left( \sum_{i=1}^{N_E} I_i \right)^2 \tag{1}$$

where $\overline{\gamma}_{s-p}$ is the average SNR, $N_E$ is the number of receive apertures, and $I_i$ denotes the receive irradiance of the link at the $i$th receive aperture.

In satellite–terrestrial laser communication, the M distribution channel model is universal for weak, moderate, and strong turbulence. The PDF for the M distribution is as follows [27]:

$$f_{I_a}(I) = A \sum_{k=1}^{\beta} a_k I^{\frac{\alpha+k}{2}-1} K_{\alpha-k} \left( 2\sqrt{\frac{I}{B}} \right) \tag{2}$$

where $a_k = \binom{\beta-1}{k-1} \frac{(g\beta+\Omega')^{1-k/2}}{(k-1)!} \left( \frac{\Omega'}{g} \right)^{k-1} \left( \frac{\alpha}{\beta} \right)^{k/2}$, $A = \frac{2\alpha^{\alpha/2}}{g^{1+\alpha/2}\Gamma(\alpha)} \left( \frac{g\beta}{g\beta+\Omega'} \right)^{\alpha/2+\beta}$, and

$B = \frac{g\beta+\Omega'}{\alpha\beta}$. $\alpha$ is a positive parameter for the effective number of large-scale cells in the scattering process, and $\beta$ is a natural number indicating the number of the fading parameters. $g = 2b_0(1-\rho)$, where $2b_0$ is the average power of the total scattered component, and $\rho$ is the ratio of the scattered power of the line-of-sight (LOS) coupling to the power of all scattered components. $\Omega' = \Omega + 2b_0 + 2\sqrt{2b\Omega\rho}\cos(\phi_A - \phi_B)$, where $\Omega$ is the average power of the LOS term, $\phi_A$ and $\phi_B$ are the determined phases of the LOS and the coupled-to-LOS scattering terms, respectively, and $K_v(\bullet)$ is the Bessel correction function of the second kind of order $v$.

Under the M-distributed channel model, the PDF of the communication system with EGC at the receiver has been given in [16]. It can be written as

$$f_{s-p}(I) = C \sum_{k=0}^{N_E(\beta-1)} c_k I^{\frac{N_E\alpha+N_E+k}{2}-1} K_{N_E\alpha-N_E-k} \left( 2\sqrt{\frac{N_E I}{B}} \right) \tag{3}$$

where $c_k = \begin{pmatrix} N_E\beta - 1 \\ N_E(\beta - 1) \end{pmatrix} \frac{1}{k!} \left( \frac{\Omega'}{(g\beta+\Omega')g} \right)^k \left( \frac{\beta}{(g\beta+\Omega')N_E\alpha} \right)^{(N_E\alpha - N_E - k)/2}$, and

$C = \frac{2(N_E\alpha)^{N_E\alpha}}{g^{N_E}(N_E\beta)_{N_E-1}\Gamma(N_E\alpha)} \left( \frac{g\beta}{g\beta+\Omega'} \right)^{N_E\beta}$.

Using $K_v(x) = \frac{1}{2}G_{0,2}^{2,0}\left[ \frac{x^2}{4} \Big| \begin{matrix} - \\ v/2, -v/2 \end{matrix} \right]$ and $\gamma = \overline{\gamma}I^2$, $G_{p,q}^{m,n}[t|_{b_1...b_m,b_{m+1}...b_q}^{a_1...a_n,a_{n+1}...a_p}]$ is the Meijer G-function. Equation (3) can be rewritten as

$$f_{s-p}(\gamma) = \frac{C}{4\sqrt{\gamma\overline{\gamma}}} \sum_{k=0}^{N_E(\beta-1)} c_k\left(\sqrt{\gamma/\overline{\gamma}}\right)^{\frac{N_E\alpha+N_E+k}{2}-1} G_{0,2}^{2,0}\left[ \frac{N_E\sqrt{\gamma/\overline{\gamma}}}{B} \Big| \begin{matrix} - \\ \frac{(N_E\alpha-N_E-k)}{2}, -\frac{(N_E\alpha-N_E-k)}{2} \end{matrix} \right] \tag{4}$$

The CDF of $\gamma$ can be calculated using $F_{s-p}(\gamma) = \int_0^\infty f_{s-p}(\gamma)d\gamma$. Substituting Equation (4) in Equation (3) and swapping the order of variables yields

$$F_{s-p}(\gamma) = \frac{C}{2} \sum_{k=0}^{N_E(\beta-1)} c_k\left(\sqrt{\gamma/\overline{\gamma}}\right)^{\frac{N_E\alpha+N_E+k}{2}} G_{1,3}^{2,1}\left[ \frac{N_E\sqrt{\gamma/\overline{\gamma}}}{B} \Big| \begin{matrix} \Delta_1 \\ \Delta_2 \end{matrix} \right] \tag{5}$$

where $\Delta_1 = 1 - \frac{(N_E\alpha+N_E+k)}{2}$, and $\Delta_2 = \frac{(N_E\alpha-N_E-k)}{2}, \frac{(N_E+k-N_E\alpha)}{2}, -\frac{(N_E\alpha+N_E+k)}{2}$.

### 2.2. HAP-to-Terrestrial Link

In the HAP-to-terrestrial link, the transmitter first transmits the optical signal to the RIS module with $N_r$ elements. Subsequently, each RIS element reflects the optical signal to the receiver. During this process, the optical signal is affected not only by atmospheric turbulence but also by pointing errors due to beam jitter and RIS jitter. Assuming that the receiver can receive the energy of all beams, the instantaneous SNR of the link can be expressed as [28]

$$\gamma = \sum_{k=1}^{N_r} \gamma_k = \overline{\gamma} \sum_{k=1}^{N_r} I_k^2 \tag{6}$$

where $\gamma_k$ is the instantaneous SNR of the k-th channel, $\overline{\gamma}$ is the average SNR, $I_k = I_a I_p$, $I_a$ is the atmospheric turbulence, and $I_p$ denotes the pointing errors.

Figure 2 shows the pointing errors $I_p$ caused by beam jitter and RIS jitter [29]. $\beta_k$ is the reflection error angle of RIS, $\theta_k' = \sqrt{\theta_{x_k}'^2 + \theta_{2_k}'^2}$ is the superimposed pointing error angle, $\theta_{x_k}'^2 = \left(1 + \frac{L_1}{L_2}\right)\theta_{x_k} + 2\beta_{x_k}$, $\theta_{y_k}'^2 = \left(1 + \frac{L_1}{L_2}\right)\theta_{y_k} + 2\beta_{y_k}$, $\{\theta_{x_k}, \theta_{y_k}\} \sim N(0, \sigma_\theta^2)$, $\{\beta_{x_k}, \beta_{y_k}\} \sim N\left(0, \sigma_\beta^2\right)$, $L_1$ is the distance from HAP to RIS, $L_2$ is the distance from the RIS to the receiver, $L = L_1 + L_2$, $\theta_{x_k}$ and $\beta_{x_k}$ are the error angle in the horizontal direction, $\theta_{y_k}$ and $\beta_{y_k}$ are the error angle in the vertical direction, and $N(0, \sigma)$ is a normal distribution with mean 0 and variance $\sigma$.

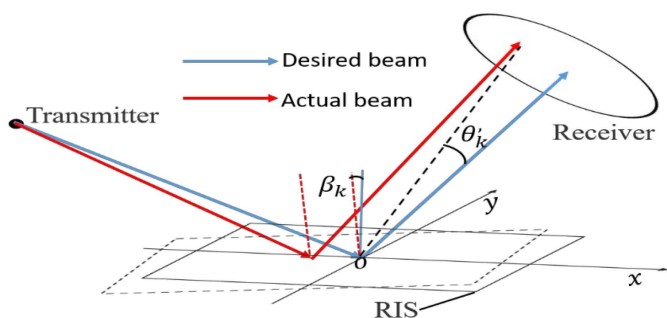

**Figure 2.** Schematic diagram of beam jitter and RIS jitter.

The PDF of $h_p$ can be expressed as [29]

$$f_{I_p}(I_p) = \frac{\xi}{A_0}\left(\frac{I_p}{A_0}\right)^{\xi-1} \tag{7}$$

where $\xi = \frac{\omega_{zeq}^2}{4\sigma_\theta^2 L^2 + 16\sigma_\beta^2 L_2^2}$, $A_0 = erfc^2(v)$, $\omega_{zeq}^2 = \omega_z^2 \frac{\sqrt{\pi}erf(v)}{2ve^{-v^2}}$, $v = \sqrt{\frac{\pi}{2}}\frac{a}{\omega_z}$, $a$ is the size of the receiving aperture, $\omega_z$ is the beam width, and $erfc(\bullet)$ is the supplementary error function.

Subsequently, the PDF of $I_k$ can be expressed as

$$f_{I_k}(I) = \int_{I|A_0}^{\infty} \frac{1}{I_a} f_{I_p}\left(\frac{I}{I_a}\right) f_{I_a}(I_a) dI_a \tag{8}$$

Substituting Equations (2) and (7) into Equation (8), and then performing a few transformations gives the PDF of $I_k^2$ as follows:

$$f_{I_k^2}(I) = \frac{A\xi}{2}\sum_{k=1}^{\beta} a_k B^{-\frac{\alpha+k}{2}} G_{1,3}^{3,0}\left[\frac{\sqrt{I}}{BA_0}\left|\begin{matrix}1+\xi\\\xi,\alpha,k\end{matrix}\right.\right] \tag{9}$$

The mean and variance of $I_k^2$ can be expressed as

$$\mu = \frac{A_0^2\xi A}{2}\sum_{k=1}^{\beta} a_k B^{2+\frac{\alpha+k}{2}} \frac{\Gamma(2+\xi)\Gamma(2+\alpha)\Gamma(2+k)}{\Gamma(3+\xi)} \tag{10}$$

$$\sigma^2 = \frac{A_0^4\xi A}{2}\sum_{k=1}^{\beta} a_k B^{4+\frac{\alpha+k}{2}} \frac{\Gamma(4+\xi)\Gamma(4+\alpha)\Gamma(4+k)}{\Gamma(5+\xi)} - \mu^2 \tag{11}$$

Using the Central Limit Theorem (CLT) [30] and assuming a large number of $N_r$, $\sum_{k=1}^{N_r} I_k^2$ can be approximated by a Gaussian random variable with mean $N_r \times \mu$ and variance $N_r \times \sigma^2$. Therefore, the PDF of $\gamma_{p-t}$ can be expressed as

$$f_{p-t}(\gamma) = \frac{1}{\sqrt{2\pi N_r \sigma^2 \overline{\gamma}}} \exp\left(-\frac{(\gamma - \overline{\gamma}N_r\mu)^2}{2\overline{\gamma}^2 N_r \sigma^2}\right) \tag{12}$$

Considering $\int_u^{\infty} \exp\left(-\frac{x^2}{4\beta} - \gamma x\right)dx = \sqrt{\pi\beta}e^{\beta\gamma^2}\left[1 - \Phi\left(\gamma\sqrt{\beta} + \frac{u}{2\sqrt{\beta}}\right)\right]$, $\Phi(\bullet)$ is the error function. The CDF of $\gamma_{p-t}$ can be obtained as

$$F_{p-t}(\gamma) = \int_0^{\gamma} f_{p-t}(\gamma)d\gamma = \frac{1}{2}\left[\Phi\left(\frac{\gamma - N_r\mu\overline{\gamma}}{\sqrt{2N_r}\overline{\gamma}\sigma}\right) - \Phi\left(\frac{-N_r\mu}{\sqrt{2N_r}\sigma}\right)\right] \tag{13}$$

## 3. Performance Analysis

In this section, we analyze the BER performance of the RIS-assisted SHTLC system. From [31], we can obtain the average BER of the DF relay system as

$$P_{BER} = P_{s-p} + P_{p-t} - 2P_{s-p}P_{p-t} \tag{14}$$

where $P_{s-p}$ and $P_{p-t}$ denote the BER of the satellite-to-HAP link and the HAP-to-terrestrial link, respectively [32].

$$P_{s-p} = \frac{q^p}{2\Gamma(p)}\int_0^{\infty} \exp(-q\gamma)\gamma^{p-1}F_{s-p}(\gamma)d\gamma \tag{15}$$

$$P_{p-t} = \frac{q^p}{2\Gamma(p)} \int_0^\infty \exp(-q\gamma)\gamma^{p-1} F_{p-t}(\gamma)d\gamma, \tag{16}$$

where $p$, $q$ are the modulation parameters for different modulation methods. For example, $p = 0.5$ and $q = 0.5$ is CBFSK; $p = 0.5$ and $q = 1$ is CBPSK; $p = 1$ and $q = 0.5$ is NBFSK; and $p = 1$ and $q = 1$ is DBPSK.

The BER of the satellite-to-HAP link is obtained as follows by transforming the exponential function in Equation (15) into the Meijer G-function and substituting it into Equation (9):

$$P_{s-p} = \frac{p^q C}{4\Gamma(p)} \sum_{k=0}^{N_E(\beta-1)} c_k \overline{\gamma}^{-\frac{N_E\alpha+N_E+k}{4}} \int_0^\infty \gamma^{\frac{N_E\alpha+N_E+k}{4}+p-1} G_{0,1}^{1,0}\left[q\gamma \Big|\begin{matrix}-\\0\end{matrix}\right] G_{1,3}^{2,1}\left[\frac{N_E\sqrt{\gamma/\overline{\gamma}}}{B}\Big|\begin{matrix}\Delta_1\\\Delta_2\end{matrix}\right] d\gamma \tag{17}$$

Using the integral property of the Meijer G-function, the closed expression for the BER of the satellite-to-HAP link is obtained as

$$P_{s-p} = \frac{C}{16\pi\Gamma(p)} \sum_{k=0}^{N_E(\beta-1)} c_k(q\overline{\gamma})^{-\frac{N_E\alpha+N_E+k}{4}} G_{3,6}^{4,3}\left[\frac{N_E^2}{16B^2q\overline{\gamma}}\Big|\begin{matrix}\Delta_3\\\Delta_4\end{matrix}\right] \tag{18}$$

where $\Delta_3 = \frac{1}{2} - \frac{N_E\alpha+N_E+k}{4}, 1 - \frac{N_E\alpha+N_E+k}{4}, 1 - p - \frac{N_E\alpha+N_E+k}{4}$, and $\Delta_4 = \frac{N_E\alpha-N_E-k}{4}, \frac{1}{2} + \frac{N_E\alpha-N_E-k}{4}, \frac{k+N_E-N_E\alpha}{4}, \frac{1}{2} + \frac{k+N_E-N_E\alpha}{4}, -\frac{N_E\alpha+N_E+k}{4}, \frac{1}{2} - \frac{N_E\alpha+N_E+k}{4}$.

The Gauss–Laguerre quadrature formula [33] can be used to transform Equation (16) into

$$\begin{aligned}P_{p-t} &= \frac{q^p}{4\Gamma(p)} \int_0^\infty \gamma^{-\frac{1}{2}} \exp(-\gamma)\exp(\gamma - \gamma q)\gamma^{p-\frac{1}{2}}\left[\Phi\left(\frac{\gamma-N_r\mu\overline{\gamma}}{\sqrt{2N_r\overline{\gamma}}\sigma}\right) - \Phi\left(\frac{-N_r\mu}{\sqrt{2N_r}\sigma}\right)\right]d\gamma \\ &= \sum_{m=1}^n H_m g(a_m)\end{aligned} \tag{19}$$

$$g(a_m) = \frac{q^p}{4\Gamma(p)} e^{a_m - qa_m} a_m^{p-0.5}\left[\Phi\left(\frac{a_m - N_r\mu\overline{\gamma}}{\sqrt{2N_r\overline{\gamma}}\sigma}\right) - \Phi\left(\frac{-N_r\mu}{\sqrt{2N_r}\sigma}\right)\right] \tag{20}$$

$$H_m = \frac{\Gamma(n+1/2)a_m}{n!(n+1)^2\left[L_n^{-0.5}(a_m)\right]^2} \tag{21}$$

where $a_m$ is the m-th root of the generalized Laguerre polynomial $L_n^{-0.5}(x)$.

Finally, the closed expression of the BER for the RIS-SHTLC system is obtained by bringing Equations (18) and (19) into Equation (14).

$$\begin{aligned}P_{BER} = \quad &\frac{C}{16\pi\Gamma(p)} \sum_{k=0}^{N_E(\beta-1)} c_k(q\overline{\gamma})^{-\frac{N_E\alpha+N_E+k}{4}} G_{3,6}^{4,3}\left[\frac{N_E^2}{16B^2q\overline{\gamma}}\Big|\begin{matrix}\Delta_3\\\Delta_4\end{matrix}\right] + \\ &\frac{q^p}{4\Gamma(p)} \sum_{m=1}^n H_m e^{a_m - qa_m} a_m^{p-0.5}\left[\Phi\left(\frac{a_m - N_r\mu\overline{\gamma}}{\sqrt{2N_r\overline{\gamma}}\sigma}\right) - \Phi\left(\frac{-N_r\mu}{\sqrt{2N_r}\sigma}\right)\right] - \\ &\frac{q^p C}{32\pi\Gamma(p)^2} \sum_{k=0}^{N_E(\beta-1)} c_k(q\overline{\gamma})^{-\frac{N_E\alpha+N_E+k}{4}} G_{3,6}^{4,3}\left[\frac{N_E^2}{16B^2q\overline{\gamma}}\Big|\begin{matrix}\Delta_3\\\Delta_4\end{matrix}\right] \times \\ &\sum_{m=1}^n H_m e^{a_m - qa_m} a_m^{p-0.5}\left[\Phi\left(\frac{a_m - N_r\mu\overline{\gamma}}{\sqrt{2N_r\overline{\gamma}}\sigma}\right) - \Phi\left(\frac{-N_r\mu}{\sqrt{2N_r}\sigma}\right)\right]\end{aligned} \tag{22}$$

## 4. Simulation Results and Analysis

Based on the above theoretical analysis and derived expressions, the performance of the RIS-SHTLC system is simulated and analyzed under the M distribution channel model, and the accuracy of the numerical results is verified using Monte Carlo simulations. Under weak turbulence, the atmospheric refractive index structure constant is taken as $7.5 \times 10^{-17}$. Table 2 shows the specific parameters of the RIS-SHTLC system.

**Table 2.** Selection of simulation parameters for RIS-SHTLC laser communication system.

| System Parameters | Symbol | Value |
|---|---|---|
| Height of Satellite | $H_s$ | 36,000 km |
| Height of receiver above ground | $h_0$ | 10 m |
| Height of HAP | $H_p$ | 20 km |
| Laser wavelength | $\lambda$ | $1550nm$ |
| Wind speed | v | $21m/s$ |
| Transmitter beam radius | $\omega_0$ | 0.1 m |
| Receiver diameter | D | 0.2 m |
| Zenith angle | $\zeta$ | $30°$ |
| The difference between the determined phase of LOS and coupled-to-LOS scattering term | $\phi_A - \phi_B$ | $\pi/2$ |
| The average power of the total scatter component | $2b_0$ | 0.2158 |
| The average power of the LOS component | $\Omega$ | 1.3265 |
| Distance from HAP to RIS | $L_1$ | 15 km |
| Distance from RIS to the ground receiver | $L_2$ | 10 km |
| Reflected error angular variance | $\sigma_\beta^2$ | 0.1 mrad |
| Pointing error angular variance | $\sigma_\theta^2$ | 0.1 mrad |

Digital modulation techniques have now become an important means to enhance the performance of satellite–terrestrial laser communication systems. Figure 3 shows the effects of CBFSK modulation, CBPSK modulation, NBFSK modulation, and DBPSK modulation on the performance of the RIS-SHTLC system, respectively. Under all four modulation schemes, the BER of the RIS-SHTLC system decreases monotonically as the average SNR increases. The BER performance of the systems that use DBPSK and CBFSK modulations is almost the same. To achieve the same BER of $10^{-5}$, the average SNR required for the RIS-SHTLC system with CBPSK modulation is 30 dB. Compared with the CBPSK modulation scheme, the system requires an average SNR of 4 dB higher with DBPSK and CBFSK modulations, and an average SNR of 10 dB higher with the NBFSK modulation. It can be observed that among the above four schemes, the RIS-SHTLC system with the CBPSK modulation method can obtain the optimal system performance. Therefore, the CBPSK modulation in RIS-SHTLC systems can achieve a better system performance compared with the CBFSK, NBFSK and DBPSK modulations.

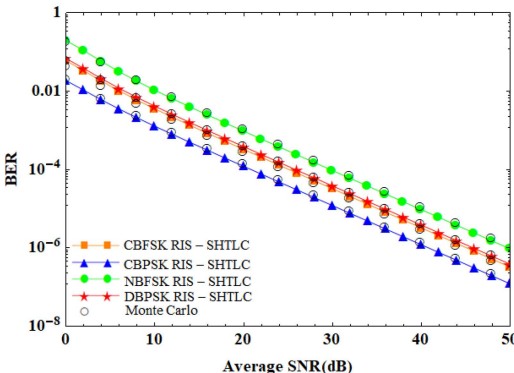

**Figure 3.** Effect of different modulation techniques on the performance of RIS-SHTLC systems.

Figure 4 shows the average SNR versus BER for SHTLC and RIS-SHTLC systems with different numbers of RIS elements. We assume $N_E = 2$ with CBPSK modulation. The results show that the BERs of all four schemes decrease as the average SNR increases. When the average SNR is small, the BER difference between SHTLC and RIS-SHTLC systems is small. The difference between them increases as the average SNR increases. The SHTLC system has a higher BER than the system with the RIS technology, and the BER of the RIS-SHTLC system can be further reduced with the addition of RIS elements. For example, the BER of SHTLC systems is $4.3 \times 10^{-4}$ at an average SNR of 30 dB. The

values of the BER of $N_r = 20, 40, 60$ RIS-SHTLC systems are $1.2 \times 10^{-5}$, $9.6 \times 10^{-7}$, and $1.2 \times 10^{-7}$, respectively. The RIS-SHTLC system with $N_r = 60$ has better BER performance. To meet the minimum requirement of BER = $10^{-5}$ for a communication system, the required average SNR of SHTLC systems under weak turbulence exceeds 50 dB. However, in the RIS-SHTLC systems with $N_r = 20$, $N_r = 40$, and $N_r = 60$ the average required SNR values are 30 dB, 20 dB, and 12 dB respectively. It can be observed that in order to achieve the same BER performance, the RIS-SHTLC system requires a lower average SNR compared to the SHTLC scheme. This trend signifies that the use of RIS technology can effectively reduce the difficulty of on-board laser temperature control and system energy consumption.

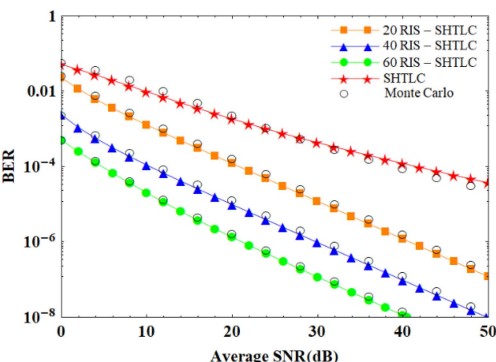

**Figure 4.** Average SNR versus BER for SHTLC systems and RIS-SHTLC systems with different numbers of RIS.

Figure 5 shows the effect of the zenith angle on the BER of the system for a different number of RIS elements. The SNR is set to 30 dB, and CBPSK modulation is used. The results show that the BER increases as the zenith angle increases, and decreases as the number of RIS elements increases. For a small zenith angle, the trend of BER variation is flatter for schemes with different numbers of RIS elements. The system BER increases rapidly and the performance of the RIS-SHTLC system decreases rapidly as the zenith angle increases. In addition, for a small zenith angle, the schemes with different numbers of RIS elements have a greater impact on the system BER, and the difference between them gradually decreases as the zenith angle increases. For example, for a zenith angle of $30°$, the BER values of RIS-SHTLC systems with $N_r = 20, 40, 60$ are $1 \times 10^{-6}$, $1.3 \times 10^{-8}$, and $1.9 \times 10^{-10}$, respectively. When the zenith angle is $70°$, the BER values of the above three schemes are $1.7 \times 10^{-5}$, $5.9 \times 10^{-6}$, and $5.4 \times 10^{-6}$, respectively. In addition, an increase in the number of RIS elements can result in a larger zenith angle for the same communication performance conditions. In satellite–terrestrial laser communication, the larger the zenith angle, the larger the communication coverage. Therefore, the use of RIS technology can optimize the performance of the SHTLC system and reduce the communication cost.

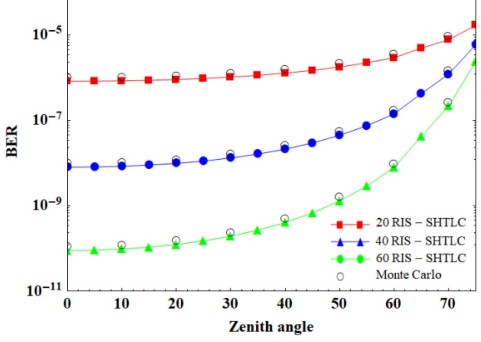

**Figure 5.** Effect of zenith angle on the performance of RIS-SHTLC system with different numbers of RIS elements.

Figure 6 shows the relationship between the receiving aperture and the BER for the RIS-SHTLC system with different RIS elements. It can be observed that at an average SNR of 40 dB in the weak turbulence case, the BER of the system with different numbers of RIS elements decreases as the receiving aperture increases. As the receiving aperture increases, the BER difference between systems with different number of RIS elements gradually increases. For a constant size of the receiving aperture, the BER of the system can be reduced effectively by increasing the number of RIS elements. To meet the same requirement of BER = $10^{-5}$ at $N_r = 20$, the RIS-SHTLC system requires a receiving aperture of 0.65 m. The required receiving apertures for RIS-SHTLC systems with $N_r = 40$ and $N_r = 60$ are 0.5 m and 0.4 m, respectively. It is obvious that as the number of RIS elements increases, the requirement for the system to achieve the same BER performance for the receiving aperture gradually becomes less strict. A small receiving aperture reduces the impact of beam distortion on system performance and reduces the transmission costs. In addition, a small receiving aperture facilitates the movement of ground terminals and increases the flexibility of the optical network.

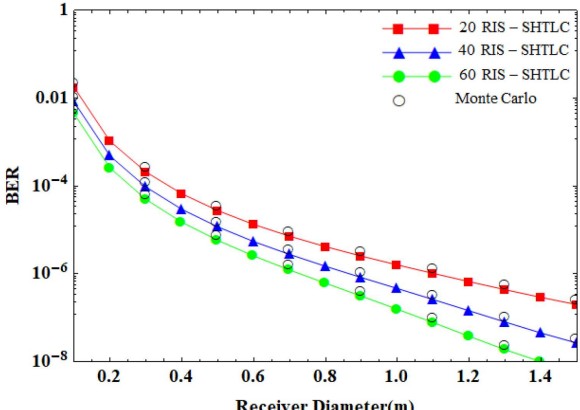

**Figure 6.** Effect of receiving aperture on the performance of RIS-SHTLC system with different numbers of RIS elements.

Figure 7 illustrates the effect of the divergence angle on the BER performance of the RIS-SHTLC system for different numbers of RIS elements. In the systems with different numbers of RIS elements, the system BER tends to increase and then decrease, and finally increase again as the divergence angle increases. When the divergence angle is less than $20 \times 10^{-5}$ rad, the system BER increases as the divergence angle increases, and increasing the number of RIS elements does not improve system performance. When the divergence angle is greater than $20 \times 10^{-5}$ rad, the BER of the system decreases and then increases as the divergence angle increases. There is an optimal divergence angle for the RIS-SHTLC system. This trend of the BER increasing and decreasing with respect to the divergence angle becomes more obvious as the number of RIS elements increases. In this scenario, the increase in the number of RIS elements can effectively improve the system BER performance while increasing the optimal divergence angle. For example, when $N_r = 20, 40, 60$, the optimal divergence angles of the corresponding RIS-SHTLC systems are $40 \times 10^{-5}$ rad, $55 \times 10^{-5}$ rad, and $60 \times 10^{-5}$ rad, respectively. In practical applications, the use of a small divergence angle requires highly accurate beam pointing control. A large divergence angle can be obtained more easily compared to a small one. The SHTLC system using the RIS technology can easily attain the optimal divergence angle; therefore, the system can achieve the optimal performance.

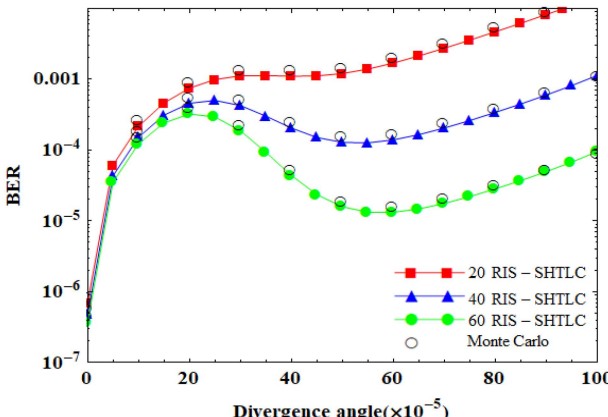

**Figure 7.** Effect of divergence angle on the performance of RIS-SHTLC system with different numbers of RIS elements.

## 5. Conclusions

This paper investigated the RIS-assisted satellite–HAP–terrestrial downlink laser communication system. The atmospheric channel was modeled using the M distribution characterization. The effect of light intensity scintillation in the satellite-to-HAP link was considered. Furthermore, the influence of both light intensity scintillation and pointing error was considered in the HAP-to-terrestrial link. Closed expressions for the BER of the RIS-SHTLC system were derived. The BER performance of the system was simulated under weak turbulence and compared with that of the SHTLC system. The effects of different modulation methods, the zenith angle, receiving aperture and divergence angle on the BER performance of the system were also investigated.

The simulation results showed that better system performance could be obtained with the RIS-SHTLC system. The BER difference between the RIS-SHTLC and SHTLC systems gradually increased with the increase in the number of RIS elements at the same average SNR. In order to meet the requirement of a minimum bit error rate of $10^{-5}$ for the communication system, the transmission power of the SHTLC system with an RIS element number of 60 is 12 dB and 8 dB lower than that of system with an RIS element number of 20 and 40 respectively. For the same communication performance, the RIS-SHTLC system could obtain better communication performance with CBPSK modulation compared to CBFSK, NBFSK, and DBPSK modulations. At the same time, when the zenith angle is 30°, the bit error rate of the SHTLC system with an RIS element number of 60 is 4 and 2 orders of magnitude lower than that of the system with an RIS element number of 20 and 40, respectively. In addition, the RIS-SHTLC system can reduce the aperture requirement of the receiver by increasing the number of RIS elements. When the system bit error rate is $10^{-5}$, the receiving aperture of SHTLC systems with 60 RIS elements is 0.15 m and 0.1 m smaller than that of systems with 20 and 40 RIS elements, respectively. When the required divergence angle was greater than $20 \times 10^{-5}$ rad, an optimal divergence angle existed that minimized the BER of the system for the RIS-SHTLC system. The optimal divergence angle became larger as the number of RIS elements increased. The optimal divergence angle of the SHTLC system with an RIS element number of 60 increases by $20 \times 10^{-5}$ rad and $5 \times 10^{-5}$ rad compared to that of the system with an RIS element number of 20 and 40, respectively. In summary, for SHTLC systems, RIS technology could effectively optimize system performance and improve the stability of the communication link. This work can provide a theoretical reference for the engineering implementation of RIS technology for satellite–terrestrial laser communication.

**Author Contributions:** Conceptualization, Y.W. and H.W.; methodology, H.W.; software, H.W.; validation, Y.W., H.W. and X.J.; formal analysis, H.W.; investigation, X.J.; resources, Y.W.; data curation, H.W.; writing—original draft preparation, H.W.; writing—review and editing, H.W.; visualization, H.W.; supervision, Y.W.; project administration, Y.W.; funding acquisition, Y.W. All authors have read and agreed to the published version of the manuscript.

**Funding:** This work was supported by Open Fund (PLN2022-06) of State Key Laboratory of Oil and Gas Reservoir Geology and Exploitation (Southwest Petroleum University) and the National Natural Science Foundation of China (Grant No. 51704267).

**Institutional Review Board Statement:** Not applicable.

**Informed Consent Statement:** Not applicable.

**Data Availability Statement:** Not applicable.

**Conflicts of Interest:** The authors declare no conflict of interest.

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
