# Peer review of "Performance of Reconfigurable-Intelligent-Surface-Assisted Satellite Quasi-Stationary Aircraft–Terrestrial Laser Communication System"

_drones, doi:10.3390/drones6120405_

Round 1

Reviewer 1 Report

Overall, the paper is written very well, the explanations are easy to follow and presented results are easy to understand. Minor remarks:

- in figure 1, instead of transmitter it should be receiver (terrestrial side)

- in table 1 some units are written italic and some not - turn italic to normal

- make a bit larger figures 3-7

Author Response

1.in figure 1, instead of transmitter it should be receiver (terrestrial side)

Author reply: Thanks for the reviewer’s kind suggestion. We have changed transmitter to receiver. The revised details are shown in Figure 1.

2.in table 1 some units are written italic and some not - turn italic to normal

Author reply: Thanks for the reviewer’s kind suggestion. We have changed all the units to non-italic. The revised details can be found in the blue part of line 231. The revised details are shown in Table 1.

3.make a bit larger figures 3-7

Author reply: Thanks for the reviewer’s kind suggestion. We have resized the figure 3-7. The revised details can be found on line 282 of the article.

Reviewer 2 Report

The English writing, presentation quality of this work is nice. Abstract and Conclusions provide a very clear idea about this work. However, I suggest Conclusion should be more precise. The mathematical model and results are comprehensively discussed. The research findings are good to be considered for publication. I have some minor suggestions to improve the quality of this work:

1.     The authors are suggested to improve the quality of figure 1.

2.     Line 22, what is SNR? Signal to Noise ratio? The authors must define acronym at the first place.

3.     Introduction section is well supported by reference literature.

4.     I suggest adding some lines about RIS along with latest research contributions will be more interesting.

5.     Authors can find latest articles reported on RIS e.g., https://www.mdpi.com/1424-8220/22/14/5278

https://www.researchgate.net/project/Reconfigurable-Intelligent-Surfaces-RIS-Intelligent-Reflecting-Surface-IRS-for-6G-wireless-communications

6.     I suggest provide a table and compare your findings with related articles.

7.     System and channel models are fine and properly discussed.

8.     I suggest check equations as some variables are not properly defined. Make it easier for readers.

9.     Why did you choose laser wavelength as 1550nm? Any certain reason for this?

10.                        Simulation results are well-explained. Findings are good.

11.                         Conclusion section must be revised. It should be more precise.

12.                        I suggest adding more references from 20220-2022 as we can find remarkable contributions in this domain during last 3 years.
